# Microbial Influences on Immune Checkpoint Inhibitor Response in Melanoma: The Interplay between Skin and Gut Microbiota

**DOI:** 10.3390/ijms24119702

**Published:** 2023-06-02

**Authors:** Youssef Bouferraa, Callie Fares, Maroun Bou Zerdan, Lucy Boyce Kennedy

**Affiliations:** 1Department of Internal Medicine, Cleveland Clinic Foundation, Cleveland, OH 44195, USA; boufery@ccf.org; 2Faculty of Medicine, American University of Beirut, Beirut 2020, Lebanon; cmf04@mail.aub.edu; 3Department of Internal Medicine, SUNY Upstate Medical University, New York, NY 13205, USA; bouzerdm@upstate.edu; 4Department of Hematology and Medical Oncology, Taussig Cancer Institute, Cleveland Clinic Foundation, Cleveland, OH 44195, USA

**Keywords:** melanoma, immune checkpoint inhibitors, gut microbiota, skin microbiota, immune-related adverse events

## Abstract

Immunotherapy has revolutionized the treatment of melanoma, but its limitations due to resistance and variable patient responses have become apparent. The microbiota, which refers to the complex ecosystem of microorganisms that inhabit the human body, has emerged as a promising area of research for its potential role in melanoma development and treatment response. Recent studies have highlighted the role of microbiota in influencing the immune system and its response to melanoma, as well as its influence on the development of immune-related adverse events associated with immunotherapy. In this article, we discuss the complex multifactorial mechanisms through which skin and gut microbiota can affect the development of melanoma including microbial metabolites, intra-tumor microbes, UV light, and the immune system. In addition, we will discuss the pre-clinical and clinical studies that have demonstrated the influence of different microbial profiles on response to immunotherapy. Additionally, we will explore the role of microbiota in the development of immune-mediated adverse events.

## 1. Introduction

Skin cancer is a major public health concern with increasing incidence rates worldwide. It has been estimated that melanoma alone will account for around 100,000 new cancer diagnosis cases in 2022 [1]. Over the past decade, immunotherapy has transformed the treatment landscape for advanced melanoma and other skin cancers [2]. Immune checkpoint inhibitors (ICIs) have demonstrated significant efficacy by unleashing the power of the immune system to recognize and attack cancer cells [2].

Despite the impressive clinical outcomes achieved with ICIs, a considerable proportion of patients fail to respond to therapy or develop resistance over time [2]. Several factors contribute to the variability in treatment response, including tumor heterogeneity, host factors, and environmental factors. Among these, the role of the gut microbiota in modulating the response to cancer immunotherapy has gained increasing attention in recent years [3]. The gut microbiota has been shown to shape the immune system and influence the efficacy of cancer immunotherapy in preclinical and clinical studies (as reviewed in [4,5,6]).

However, the role of skin microbiota in the pathogenesis of melanoma and their response to ICIs remains largely unexplored. The skin microbiota is a diverse community of microorganisms that inhabit the skin surface and play a critical role in maintaining skin homeostasis and host defense [7].

Understanding the contribution of the skin and gut microbiota to the response of skin cancers to ICIs is essential for improving treatment outcomes and developing personalized strategies for cancer immunotherapy. In our previous work published in 2021, we discussed the role that the gut microbiota plays in overcoming the resistance to ICIs used in the treatment of different cancer types [8]. As the field has significantly evolved in the past couple of years, the role of gut microbiota has been extensively studied in many types of cancer, requiring an extensive review for its role in every cancer type alone.

In this article, we aim to discuss the potential mechanisms underlying the interaction between the skin and gut microbiota and the immune system in melanoma, the current evidence supporting the role of the microbiota in ICI response, and the future implications for clinical practice. As such, this article will focus on the role of gut as well as skin microbiota in the development and response of melanoma to ICIs, providing a focused update on this interaction in melanoma patients in particular [8].

## 2. Role of Skin and Gut Microbiota in the Development of Melanoma

### 2.1. Proposed Mechanisms of Skin Microbiota Influence on Melanoma Development

Many kinds of skin commensal bacteria have been shown to promote skin immunity and consequently protect against skin infections, inflammatory disorders, and malignancies [9,10,11,12]. On the other hand, some chronic skin conditions may lead to alterations in the skin microbiome. Although no causative bacterial pathogen has been identified in melanomagenesis, alterations of the skin microbiome in chronic skin conditions may lead to colonization with pathogenic bacteria, which may in turn play a role in the development of non-melanoma skin cancers [13]. In addition, some small studies have shown differences in the skin microbiota in melanomas [14,15]. In a study by Mizuhashi et al., the presence of Corynebacterium was found to be much higher in patients with stage III/IV melanoma (76.9%) compared to those with stage I/II melanomas (28.6%) [14]. In addition, Salava et al. reported a numerically decreased skin microbial diversity in melanomas as compared to benign nevi; however, the results were not statically significant [15].

The skin immune system is composed of the innate and adaptive immune system. Keratinocytes constitute an essential component of the innate immune system and produce a variety of chemokines, cytokines, and antimicrobial peptides [16]. Antimicrobial peptide production is controlled and upregulated by microbial stimuli known as microorganism-associated molecular patterns (MAMPs) [17]. In this way, skin microbial flora regulate the innate immune system. Antimicrobial peptides bind to pattern recognition receptors (PRRs) located on keratinocytes, antigen-presenting cells, and melanocytes orchestrating the immune response [18]. PRRs include intracellular cytoplasmic receptors and Toll-like receptors TLRs [19]. Persistent activation of TLRs has been implicated in chronic inflammation and skin carcinogenesis [20].

Skin barrier disruption can alter microbial homeostasis and lead to microbial dysbiosis [21]. In addition, skin microbes can contribute to barrier dysfunction by releasing proteases that damage the epidermal lining [22]. While microbiota-specific barrier disruption has not yet been proven to promote skin cancers, skin barrier disruption in general and consequent chronic inflammation has been related to non-melanoma skin cancers in multiple studies [22,23]. It also remains unclear whether the skin barrier disruption and consequent microbial dysbiosis or the microbial-induced skin barrier damage constitute the inciting event of chronic inflammation and carcinogenesis.

Moreover, it has been suggested that ultraviolet radiation (UV)-induced immunosuppression and consequent skin carcinogenesis can be inhibited by skin microbiota through alteration of cytokine gene expression and immune cell infiltration in the skin [24]. In a study by Patra et al. using germ-free mice, epidermal hyperplasia and neutrophil infiltration in UV exposed skin were higher in the presence of skin microbiota, whereas mast cells, macrophages, and monocytes were more prominent in the absence of microbes [24]. In addition, genetic expression of proinflammatory cytokines was higher in colonized skin, whereas increased expression of immunosuppressive cytokines was observed in their germ-free counterparts [24]. Other studies have reported a protective effect of lactobacilli against UV-induced skin carcinogenesis [25,26].

Overall, intra-tumoral microbes can impact skin carcinogenesis by directly interacting with cancer cells or regulating other components of the tumor microenvironment (TME). Intra-tumoral bacterial composition in melanoma may influence the extent of immune cell infiltration, chemokine expression, and overall prognosis. For example, in a study using RNA sequencing data from The Cancer Genome Atlas in cutaneous melanoma, patients with low levels of intratumoral CD8+ T cells had significantly shorter survival compared to those with high levels. In this study, intratumoral bacterial load of *Lachnoclostridium* was positively associated with infiltrating CD8+ T cells, suggesting that intratumoral microbiota may affect intratumoral immune cell infiltration, thereby influencing survival [27]. To date, strategies to modulate the skin microbiome have not been studied as a therapeutic intervention for melanoma. There is a prospective study, SKINBIOTA (NCT 04734704), which will analyze the composition of the skin microbiota using skin swabs in patients treated with anti-PD-1 for metastatic melanoma. This study will contribute to the emerging field examining the interplay between composition of the skin microbiome and immunotherapy response and resistance.

### 2.2. Role of Gut Microbiota in the Development of Skin Cancers: Effects on the Immune System

The gut microbiota has also been shown to play a role in the development of skin cancers [28]. In fact, gut microbiota have been shown to have both oncogenic and tumor-suppressive properties that can exert specific effects on multiple types of cancers, including skin cancers [28]. In a study by Luo et al., *Lactobacillus reuteri* FLRE5K1was shown to stimulate the production of anti-oncogenic cytokines in mice and prevent the migration of melanoma cells, thereby preventing the development of melanoma and prolonging survival [29]. In another pre-clinical study, supplementation with VSL#3 probiotics was found to trigger the production of butyrate and propionate by gut microbiota [30]. This cascade led to the recruitment of Th17 cells, which in turn reduced lung metastases and decreased the number of tumor foci [30]. Li et al. demonstrated that transferring 11 bacterial strains, which were more abundant in mice with negative ubiquitin ligase RNF5, resulted in the development of anti-tumor immunity and limited melanoma growth in germ-free mice [31].

On the contrary, other studies have shown that gut microbiota can promote oncogenesis in skin cancers [32]. Gut bacterial profiles have been shown to be significantly different between melanoma and control patients, with changes in the bacterial composition with progression from in situ to invasive and, later, metastatic melanoma [32]. In particular, *Saccharomytecales* and *Prevotella copri* species were more abundant in advanced stages of melanoma [32]. Furthermore, Pereira et al. found that IL-6 and the microbiota of obese mice can promote the advancement of melanoma [33]. They conducted fecal transplants experiments using leptin-deficient mice and found that the transfer resulted in tumor development in lean mice [33]. In addition, microbial depletion using oral antibiotics leads to reduced burden of subcutaneous and hepatic melanoma in mice, indicating a potential role of the gut microbiota in the progression of melanoma [34]. All these studies suggest that interventions targeting the gut microbiota constitute potential therapeutic modalities to target the development and progression of melanoma.

## 3. Role of the Microbiota in Influencing Response to Immune Checkpoint Inhibitors (ICIs)

### 3.1. Pre-Clinical and Clinical Studies Studying the Microbial Profiles’ Influence on the Response to ICIs

Immune checkpoint inhibitors (ICIs) represent a significant advance in the field of cancer immunotherapy and are widely used across multiple tumor types. These drugs specifically target immune checkpoints, including programmed cell death 1 (PD-1), PD ligand 1 (PD-L1), and cytotoxic T-cell lymphocyte-associated protein (CTLA-4) (Figure 1) [8]. Immune checkpoints are a complex set of stimulatory and inhibitory proteins that play a crucial role in regulating the T-cell immune response. They are responsible for controlling the activation of cytotoxic T-lymphocytes, maintaining self-tolerance, preventing autoimmunity, and adjusting the duration and strength of the immune response to minimize tissue damage during inflammation [35,36]. Several preclinical and clinical studies have shown that the responsiveness of multiple cancer types to immune checkpoint inhibitors (ICIs) relies on the microbiota present in the gut and the skin.

In a study by Routy et al., the therapeutic efficacy of anti-PD-1 alone or in combination with anti-CTLA-4 was compared between antibiotic-treated (germ-free) or untreated mice with melanoma [3]. The administration of antibiotics had a considerable negative impact on the efficacy of anti-PD-1 monoclonal antibody therapy, either alone or in combination with anti-CTLA-4 antibodies, resulting in increased tumor size, reduced antitumor effects and decreased survival in germ-free mice [3]. In addition, colonizing the intestines of germ-free mice with fecal transplants rich in *Akkermansia muciniphilia* restored the responsiveness of melanoma-bearing hosts to ICIs, a response that had been previously inhibited with the use of antibiotics [3]. Similar results were also reported in another study using mice with melanoma treated with anti-CTLA-4 antibodies [37]. Vetizou et al. showed that antibiotic-treated mice with melanoma did not respond to anti-CTLA-4, until colonized with *Bacteroides fragilis* [37]. Oral supplementation with *B. fragilis* in germ-free mice restored the therapeutic response of anti-CTLA-4 via the induction of T helper 1 (TH_1_) immune responses in tumor-draining lymph nodes (LN) and the promotion of the maturation of intra-tumoral dendritic cells (DC) [37]. In addition, germ-free mice with melanoma that received fecal microbiota transplant (FMT) from melanoma patients with a strong response to CTLA-4 had better outcomes after treatment with ICIs as compared to those with FMT from non-responder patients, with the former group favoring the growth of *B. fragilis* [37].

A study by Gopalakrishnan et al. examined the feces of 112 patients with melanoma treated with anti-PD-1 therapy. The patients’ gut microbiota was examined pre- and post-treatment via 16S sequencing and metagenomic whole genome shotgun sequencing. Patients with a more diverse gut microbiome had a better response to anti-PD-1 therapy compared to patients with less diverse gut microbiome. The microbiota of responding patients were enriched with the *Clostridiales* order, the *Ruminococcaceae* family and the *Faecalibacterium* genus, whereas those of non-responding patients were enriched with *Bacteroidales* [38]. Similar findings in patients who received anti-CTLA-4 have been demonstrated. In fact, in a prospective study of patients who received ipilimumab for metastatic melanoma, patients whose baseline gut microbiota was enriched for *Faecalibacterium* had longer progression-free survival versus patients whose gut microbiota was enriched for *Bacteroidale* [39]. However, this is opposed to the findings by Vetizou et al. summarized above, where *Bacteroidales* was associated with better response to anti-CTLA-4 [37]. Gopalakrishnan et al. also performed FMT from the responding patients and non-responding patients into mice. Mice transplanted with responding FMT had better response to anti-PD-L1 therapy. These mice were found to have a higher abundance of *Faecalibacterium* in their gut microbiota [38].

Finally, by studying the response of 38 patients with metastatic melanoma to anti-PD-1 and anti-CTLA-4, Andrews et al. showed that responders had different gut microbial composition compared to non-responders. Through 16S rRNA gene sequencing and shotgun metagenome sequencing of fecal samples, they showed that patients who were more likely to respond had microbiome rich in *Bacteroides stercoris*, *Parabacteroides distasonis*, and *Fournierella massiliensis.* However, non-responding patients were more likely to have a microbial composition rich in *Klebsiella aerogenes* and *Lactobacillus rogosae* [6]. In order to address discrepancies between different studies, McCulloch et al. assessed the microbial composition of five different melanoma cohorts [40]. In their study, time-to-event analysis revealed that the baseline microbiota composition was optimally linked with clinical outcome after about one year of treatment initiation [40]. When the combined data were analyzed through meta-analysis and other bioinformatic methods, it was found that the *Actinobacteria* phylum and the *Lachnospiraceae/Ruminococcaceae* families of Firmicutes were associated with a favorable response, whereas Gram-negative bacteria were associated with an inflammatory intestinal gene signature, increased blood neutrophil-to-lymphocyte ratio, and unfavorable outcome [40]. Two microbial signatures, one enriched for *Lachnospiraceae* spp. and the other for *Streptococcaceae* spp., were linked with favorable and unfavorable clinical response, respectively, and with distinct immune-related adverse effects [40]. Despite variations between different cohorts, optimized learning algorithms that were trained on batch-corrected microbiome data consistently predicted outcomes for programmed cell death protein-1 therapy in all cohorts [40]. In summary, these studies examining the gut microbiome of immunotherapy responders and non-responders demonstrate an association between the gut microbiome and ICI response and resistance. Taken together these findings suggest that the gut microbiome is an exciting therapeutic target to overcome ICI resistance. However, conflicting results have been published regarding the prognostic impact of specific microbial signatures, and uncertainty remains regarding the optimal evaluation and interpretation of the gut microbiome as a biomarker of immunotherapy response and toxicity. Additional large-scale studies are needed to determine if specific microbial profiles can be linked to ICI response and resistance and clinically used as biomarkers.

Complete response to anti-PD-1 antibodies occurs in 10–20% of patients with metastatic melanoma, and the majority of patients who receive anti-PD-1 antibodies for metastatic melanoma will ultimately develop resistance [41]. There is an unmet need to identify biomarkers to predict ICI resistance and to develop novel treatment strategies to overcome ICI resistance [42]. One proposed strategy to overcome ICI resistance relies on using FMT. This technique requires transplantation of donor fecal matter into the recipient’s intestinal tract, facilitating a transformation in the recipient’s microbial composition [42,43]. Two phase 1 trials were recently published in *Science* investigating FMT from immunotherapy responders combined with anti-PD-1 antibodies as a strategy to overcome anti-PD-1 resistance in patients with metastatic melanoma [42,44]. Baruch et al. performed a phase I clinical trial to study feasibility, safety, and immune cell impact of FMT combined with the reintroduction of anti-PD-1 for patients with anti-PD-1 refractory melanoma [42]. Two FMT donors were chosen based on complete response to previous anti-PD-1 monotherapy for metastatic melanoma [42]. There were ten total recipients, who received equal transplantation from each donor. Three recipients showed a response to anti-PD-1 treatment, all from a similar donor (Donor #1), including one patient with complete response (CR) and two patients with partial response (PR). Stool 16S gene sequencing analysis showed a significant difference between pre- and post- treatment microbiota of the recipients. The post-treatment microbiota also differed between the recipients of the different donors. The recipient groups from donor #1 contained a higher abundance of *Bifidobacterium adolescentis*, whereas the recipients from Donor #2 had a higher abundance of taxa like *Ruminococcus bromii.* Further analysis found that responders had a higher relative abundance of *Enterococcaceae*, *Enterococcus*, and *Streptococcus australis* and a lower relative abundance of *Veillonella atypica*. Interestingly, only recipients from donor #1 upregulated some additional gene sets related to APCs activity, innate immunity, and interleukin-12 (IL-12), with the responding patients increasing the CD8+ T cell infiltration into the tumors [42]. In addition, Davar et al. showed that in patients with advanced melanoma who were resistant to anti-PD-1 therapy, the combination of responder-derived FMT and anti-PD-1 was found to be safe and effective [44]. Out of 15 patients, 6 experienced clinical benefit including 1 patient with CR, 2 patients with PR, and 3 patients with SD lasting > 12 months, and the microbiota was perturbed rapidly and durably [44]. The responders showed an increase in the abundance of Firmicutes (*Lachnospiraceae* and *Ruminococcaceae* families) and Actinobacteria (*Bifidobacteriaceae* and *Coriobacteriaceae* families) taxa that were previously linked to the response to anti-PD-1, as well as increased activation of CD8+ T cells and a decrease in the frequency of myeloid cells expressing interleukin-8 [44]. Additionally, the responders had distinct proteomic and metabolomic signatures, and the gut microbiome was shown to regulate these changes through transkingdom network analyses [44].

While it can be difficult to draw definite conclusions regarding the effect of specific microbial species on the response to ICIs given the inconsistencies between different studies, it has been noticed that *Akkermansia muciniphilia*, *Bacteroide fragilis*, and *Fecalibacterium* tend to be usually associated with a positive outcome, unlike *Bacteroidales* which usually negatively impact the response of melanoma patients to ICIs. It is worth noting, however, that microbial diversity remains the only consistent finding associated with positive response outcomes along the different studies.

Table 1 and Table 2 summarize the main published preclinical and clinical studies assessing the gut microbiota and response to ICIs. Despite all the above, many challenges still exist when it comes to drawing clinical conclusions from the above data. In fact, most studies cited above had small sample sizes. The majority are also non-randomized single arm early phase trials. In addition, as noted above, many studies had discordant data, and no specific microbial species has been consistently associated with a positive or negative response to ICIs in melanoma patients. That being said, several clinical trials are currently running to broaden our clinical understanding of this complex interaction. Table 3 and Table 4 summarize the ongoing clinical trials and observational studies currently assessing the interaction between the microbiota and response to ICIs in melanoma patients.

### 3.2. Proposed Mechanisms through Which Microbiota Influences the Response to ICIs

The mechanisms through which the microbiota influences the response to ICIs have been extensively studied. Some of those potential mechanisms are summarized in Figure 2.

The impact of the microbiota on anti-tumor immune cell infiltration is the most studied mechanism of interaction between the microbiota and the response to ICIs. At the level of the tumor microenvironment, a higher density of CD8+ T cells were observed in responding patients as compared to non-responding patients. This CD8+ T cell infiltration was positively correlated with the *Clostridiales* order, the *Ruminococcaceae* family, and the *Faecalibacterium genus*, and it was non-significantly but negatively correlated with Bacteroidales. In addition, a higher level of systemically circulating effector CD4+ and CD8+ T cells with a preserved cytokine response to anti-PD-1 therapy was associated with *Clostridiales* order, the *Ruminococcaceae* family, and the *Faecalibacterium*. On the other hand, gut microbiota enriched with Bacteroidales was associated with higher levels of Treg cells and myeloid-derived suppressor cells in the systemic circulation with a blunted cytokine response to anti-PD-1 therapy [38]. Moreover, and similarly to what was found in human patients, mice transplanted with responder FMT were also found to have a higher density of CD8+ T cells. Furthermore, an upregulation of PD-L1 was also established in the mice models. In fact, mice receiving FMT from responders were also found to have a higher frequency of innate effector cells and a lower frequency of suppressive myeloid cells [38]. Additionally, studies in mouse models have discovered that colonizing germ-free mice with specific gut microbiota leads to an increase in CD8+ T cell infiltration into the tumors and an increase in CD8+ and CXCR3+ CD4+ T cells in the circulation [45]. This, in turn, results in an increase in type 1 immune response [45]. Similarly, Baruch et al. demonstrated an increase in intra-tumoral CD8+ T cell infiltration in FMT recipients responding to anti-PD-1. The FMT caused an increase in CD68+ APCs infiltration into the gut lamina propria [42]. Another similar study showed that responding recipients shifted closer to the donors’ gut microbiota as compared to the non-responders [44]. Increases in cytolytic CD56+ CD8+ T cells and terminally differentiated effector memory CD8+ T cells (CCR7− CD45RA+) were also noted through longitudinal single cell analyses of peripheral mononuclear blood cells and tumor-infiltrating immune cells. Regulatory T cells (Tregs) were also found to be decreased in the responding recipients [44]. Therefore, through the use of FMT, some recipients are able to respond to immunotherapy via similar mechanisms of the responding donor [45]. It is important to note, however, that research is needed to fully understand the mechanisms by which these microorganisms influence the response to ICIs.

Another proposed mechanism includes immune modulation by bacterial metabolites. Bacterial fermentation of carbohydrates into short chain fatty acids (SCFAs) has been correlated with the host immunity to ICIs [46]. The effects of butyrate have been highly studied and have shown differential influences the response to anti-PD-1 and anti-CLA4 therapy. *Faecalibacterium prausnitzii* is considered a major contributor to butyrate formation. SCFA formation has been correlated with improvement in response to anti-PD-1 therapy [39,47,48]. However, it has been shown to blunt the response with anti-CLA4 therapy in melanoma patients [49].

In addition to the production of SCFA, gut microbiota may also induce molecular mimicry to host cell antigens. Self-reactive T cells are mostly eliminated during development; however, some are able to escape [50]. These may be activated by microbial antigens that have immunogenic properties that are similar to host cell antigens [51]. Some tumor cells express self- or neo-antigens that can be recognized by the self-reactive T cells [52]. As such, this cross-reactivity has been proven to enhance the response to ICIs, mostly by T cell-mediated killing [3,45,51].

One suggested mechanism also involves the formation of anabolic amino acids by the gut microbiota. These were found to be predominant in responding patients, while catabolic amino acids were mostly predominant in non-responding patients [38]. In fact, the biosynthesis of amino acids is proposed to stimulate host immunity [38].

Furthermore, inosine, a purine riboside, has been shown to be correlated with response to ICIs in mice. Inosine is an intestinal metabolite produced by *Bifidobacterium* and *Akermansia muciniphilia*. It has been shown to enhance TH_1_ differentiation and adenosine A_2A_ receptor expressing naïve T cell function. Inosine modulates the response to ICIs by inhibiting the immunosuppressive activity of adenosine, which is a naturally occurring molecule that suppresses the immune response. In contrast to adenosine, inosine has pro-inflammatory effects on the adenosine A_2A_ receptor, supporting the TH_1_ and its anti-tumor effects in the mice [53]. Inosine mainly acts as a competitive inhibitor of adenosine by blocking its binding to its receptors and facilitating an increased immune response against cancer cells [54].

## 4. Role of Skin and Gut Microbiota in the Development of Toxicities Associated with ICIs

### 4.1. Definition of Immune-Related Adverse Event (irAE): Limitation to Using ICIs

Although use of ICIs has transformed the landscape of cancer treatment by facilitating the harnessing of the immune system to generate an anti-tumor immune response, there exist several limitations to using these drugs. First, not all patients respond to ICI, and response rates vary by tumor type [42]. In addition, ICI are expensive and may not be accessible to all patients [55]. Most importantly, ICIs can cause severe immune-related adverse events (irAEs) that can be life-threatening if not treated promptly and appropriately [56,57,58]. By definition, irAE occurs as a result of an ICI-induced “inappropriate” immune system activation against the hosts’ own cells [8]. While cutaneous irAEs are among the most common, any organ system can be involved, and side effects can range from colitis to dermatitis, hepatitis, pneumonitis, as well as endocrinopathies such as thyroiditis and hypophysitis [56,57,58]. In addition, ICIs have been associated with musculoskeletal adverse events including inflammatory arthritis, myositis, and polymyalgia rheumatica [59]. While neurotoxicity, cardiotoxicity, and pulmonary toxicity are less frequent, they tend to be the most severe and life-threatening. In addition, much remains unknown about the long-term effects of ICIs on patients and the management of irAEs [56,57,58].

### 4.2. The Role of the Microbiota in Influencing the Rate of Immune-Mediated Adverse Events

Cutaneous irAEs can range from mild pruritus to life-threatening epidermal necrolysis [56,57,58]. Hu et al. studied the effect of skin microbiota on a mouse model of cutaneous irAE [56]. Treatment with anti-CTLA-4 alone did not produce any skin inflammation in the mouse model, nor did local skin colonization with *Staphylococcus epidermidis* [56]. However, when the mice received concurrent cutaneous colonization with *S. epidermidis* and systemic anti-CTLA-4, skin inflammation developed on days 6 to 8 of treatment [56]. The inflammatory infiltrate consisted of macrophages and cytokine-producing neutrophils and monocytes [56]. This innate, hyperactive immune response to anti-CTLA-4 treatment was found to be dependent on IL-17 production by commensal-specific T cells in an excessive, dysregulated manner [56]. These findings suggest that alterations in the skin microbiome may affect development of cutaneous irAE.

Colitis is a common irAE seen with both anti-CTLA-4 and anti-PD-1 antibodies and may be severe or life-threatening [60,61]. The impact of the gut microbiome on development of immune-related colitis has been extensively studied. The *Bacteroidetes* phylum has been associated with increased resistance to colitis. Dubin et al. analyzed the composition of the intestinal microbiota in 34 patients with metastatic melanoma being treated with anti-CTLA-4. Patients who were not diagnosed with gastrointestinal inflammation between 13 and 59 days of treatment were found to have a higher abundance of *Bacteroidaceae, Rikenellaceae*, and *Barnesiellaceae* [60]. It is thought that the *Bacteroidetes* phylum stimulates Treg cell differentiation, which may play a role in certain patients’ resistance to colitis [60,62,63]. Another study by Chaput et al. showed similar findings. In this study, 26 patients with metastatic melanoma received anti-CTLA-4 and were closely observed for the development of colitis [39]. Abundance of the *Bacteroidetes* phylum was associated with resistance to colitis. In addition, patients with *Firmicutes*-rich microbiota were more likely to develop colitis. Interestingly, it was also found that decreased bacterial diversity was also associated with gastrointestinal inflammation [39].

However, opposing results were found in patients treated with dual ICIs (anti-CTLA-4 and anti-PD-1) [6]. In fact, patients that were more resistant to colitis had a higher abundance of *Firmicutes*, while patients prone to colitis had a higher abundance of *Bacteroidetes* [6]. Patients and pre-clinical models that had *Bacteroidetes*-rich profiles and that developed colitis were found to upregulate IL-1ß. This was confirmed via treatment with IL-1 receptor antagonist (anakinra), along with the dual ICIs, resulting in less inflammation. In addition, via transcriptional profiling, a prompt and selective transcriptional upregulation of *Il1b* was also found [6].

Liu et al. found a link between the composition of a patient’s gut microbiome and their likelihood of developing irAE from anti-PD-1 antibodies [64]. Patients with a less diverse gut microbiome had a higher risk of experiencing irAE [64]. A total of 150 patients were included in the study, and irAEs due to anti-PD-1 included pruritis and/or rash, thyroid dysfunction, and mild to severe diarrhea. Patients were grouped into no/mild irAE versus severe irAE group based on a National Cancer Institute Common Terminology Criteria for Adverse Events (CTCAE V5.0) grading system. Patients experiencing severe irAEs were found to have gut microbiota abundant with *Streptococcus*, *Paecalibacterium*, and *Stenotrophomonas*. However, patients with mild irAE had gut microbiota enriched for *Faecalibacterium.* Patients experiencing each irAE were analyzed and compared to patients with no irAE. The microbiota in patients experiencing pruritis and rash had no significant difference when compared to those without irAE. However, patients with no irAE had a higher abundance of *Bacteroides* and *Lactobacillus* as compared to patients experiencing thyroid dysfunction, who had abundant *Paecalibacterium*. It was also found that patients with severe diarrhea had a higher presence of *Stenotrophomonas* and *Streptococcus*, whereas patients without irAEs or with mild diarrhea had higher levels of *Faecalibacterium* and *Bacteroides* [64].

In summary, these studies highlight the interplay between the gut microbiota and development of irAE. A retrospective analysis of 327 cancer patients treated with ICI for multiple tumor types found that patients who developed diarrhea or colitis had improved overall survival compared to those who did not develop diarrhea or colitis [65]. The mechanism underlying improved survival in patients who develop immune-related colitis is not known, but it is possible that the gut microbiome may play a role and could be targeted in future prospective studies. Future work is also needed to clarify specific microbial profiles affecting colitis risk, which will improve risk assessment for and management of immune-related colitis. There are multiple ongoing, prospective trials investigating the impact of the microbiome on ICI efficacy and toxicity (NCT03643289 and NCT04107168).

Table 5 summarizes the role of microbiota in influencing the irAEs in skin cancer.

## 5. Conclusions

In conclusion, emerging evidence suggests that the composition and diversity of the skin and gut microbiota play a critical role in modulating the efficacy of immune checkpoint inhibitors in the treatment of skin cancers. Despite the progress made in the field, several challenges remain in harnessing the potential of microbiota-based therapies to optimize immune checkpoint inhibitor efficacy. Future research efforts should aim to identify specific microbial profiles that predict the response to therapy and elucidate the molecular mechanisms underlying their effects on tumor immunity. This knowledge may enable the development of microbiota-based interventions, such as fecal microbiota transplantation or probiotics, to enhance the clinical efficacy and safety of immune checkpoint inhibitors in patients with skin cancers.

## Figures and Tables

**Figure 1 ijms-24-09702-f001:**
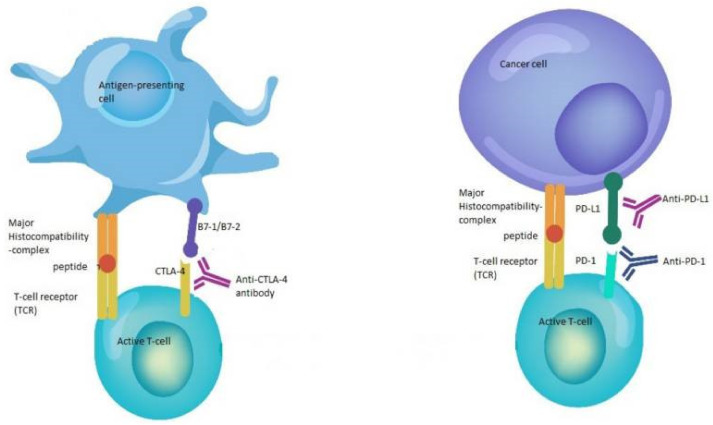
Mode of action of immune checkpoint inhibitors [8]. Immune checkpoint inhibitors work by targeting specific mechanisms that prevent T cells from attacking cancer cells in the body. One such mechanism involves the binding of B7-1/B7-2 to CTLA-4, which keeps T cells inactive and unable to kill cancer cells. Anti-CTLA-4 antibodies block this binding, enabling the T cells to become active and attack cancer cells. Another mechanism involves the binding of PD-L1 to PD-1, which also prevents T cells from attacking cancer cells. Anti-PD-1/PD-L1 antibodies interrupt this binding and enhance the ability of T cells to target and kill cancer cells.

**Figure 2 ijms-24-09702-f002:**
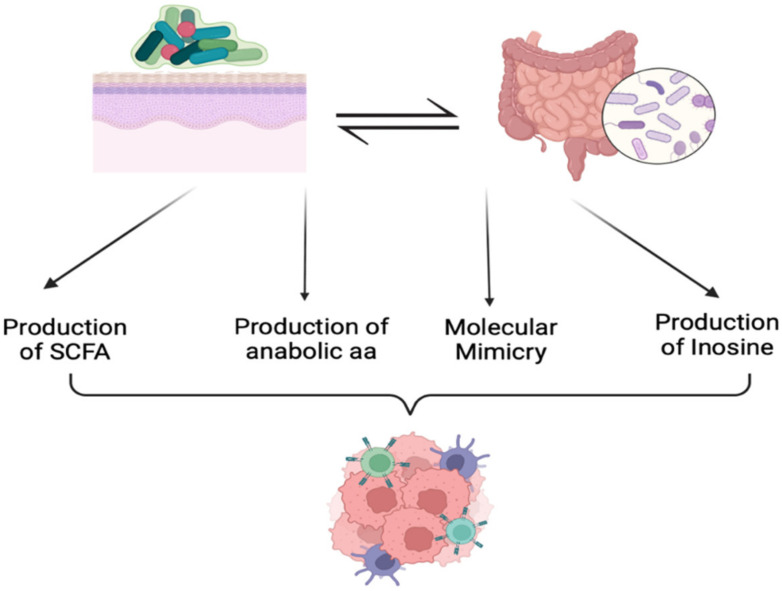
Proposed mechanisms of microbial influence on the response to immune checkpoint-inhibitors. These include production of inosine, anabolic amino acids, short chain fatty acids, as well as molecular mimicry between microbial and self-antigen. Through these mechanisms, microbiota can affect the immune cell infiltration into the tumor cells and consequent responses to immune-checkpoint inhibitors.

**Table 1 ijms-24-09702-t001:** Preclinical studies assessing the interaction between the gut microbiota and the response to immune checkpoint inhibitors in Melanoma.

Bacteria	Immunotherapy	Target Tumor	Main Findings	Reference
*Akkermansia muciniphilia*	PD-1 mAb ± CTLA-4 mAb	MCA-205, LLC and RET tumor-bearing mice	Resistance to response to ICI therapy in antibiotic-pretreated mice. Restored response with FMT from ICI responders due to increased levels of *Akkermansia muciniphila*	[3]
*Bacteroides fraglis*	CTLA-4 mAb	Melanoma	Resistance to response to ICI therapy in antibiotics-treated mice. Restored response of anti-tumor response via oral feeding with *B. fragiles.* Enhanced response with FMT from patients with increased *Bacteroides* spp. levels.	[37]
*Faecalibacterium*	PD-1 mAb	Melanoma	Increased levels of *Faecalibacterium* led to reduced tumor size and improvement in response to ICI therapy.	[38]

**Table 2 ijms-24-09702-t002:** Clinical studies assessing the gut microbiota and the response to immune checkpoint inhibitors in Melanoma.

Bacteria	Immunotherapy	Target Tumor	Main Findings	Reference
*Bifidobacterium adolescentis*, *Ruminococcus bromii*	PD-1 mAb	Melanoma (anti-PD-1-refractory metastatic melanoma)	Increased levels of *Bifidobacterium adolescentis* in ICI responders (Donor #1). Increased levels of *Ruminococcus bromii* in ICI non-responders (Donor #2).	[42]
*Enterococcaceae*, *Enterococcus*, *Streptococcus australis*, *Veillonella atypica*	PD-1 mAb	Melanoma (anti-PD-1-refractory metastatic melanoma)	Responders from Donor #1 had a higher relative abundance of *Enterococcaceae*, *Enterococcus*, and *Streptococcus australis*, and a lower relative abundance of *Veillonella atypica.*	[42]
*Clostridiales* order, *Ruminococcaceae* family, *Faecalibacterium* genus, *Bacteroidales*	PD-1 mAb	Melanoma	Increased levels of *Clostridiales* order, the *Ruminococcaceae* family and the *Faecalibacterium* genus in ICI responders. Increased levels of *Bacteroidales* in ICI non-responders.	[38]
*Faecalibacterium, Bacteroidale*	CTLA-4 mAb	Metastatic Melanoma	Increased levels of *Faecalibacterium* had longer progression-free survival versus patients with a higher abundance of *Bacteroidale.*	[39]
*Bacteroides stercoris*, *Parabacteroides distasonis*, and *Fournierella massiliensis; Klebsiella aerogenes* and *Lactobacillus rogosae*	PD-1 mAb ± CTLA-4 mAb	Metastatic Melanoma	Increased levels of *Bacteroides stercoris*, *Parabacteroides distasonis*, and *Fournierella massiliensis* in ICI responders. Increased levels of *Klebsiella aerogenes* and *Lactobacillus rogosae* in ICI non-responders.	[6]
Firmicutes (*Lachnospiraceae, Ruminococcaceae*), Actinobacteria (Bifidobacteriaceae, Coriobacteriaceae)	PD-1 mAB	Advanced Melanoma resistant to anti-PD-1	FMT from previous anti-PD-1 responders increased Firmicutes and Actinobateria in previous non-responders. Increased CD8+ T cell activation Decreased IL-8-producing myeloid cells	[44]
*Actinobacteria*, the *Lachnospiraceae/Ruminococcaceae*	PD-1 mAB	PD-1 treated melanoma	Association with decreased progression	[40]
*Bacteroides genus and* Proteobacteria	PD-1 mAB	PD-1 treated melanoma	Association with increased progression	[40]

**Table 3 ijms-24-09702-t003:** Ongoing clinical trials assessing the interaction between ICIs and microbiota in melanoma.

NCT	Trial Phase or Type	Patient Selection	Intervention or Treatment	Number of Cases
NCT05273255	Pilot	ICI-refractory melanoma	FMT (via endoscopy)	30
NCT03341143	Phase I	Anti-PD1 refractory melanoma	FMT (via colonoscopy) + pembrolizumab	16
NCT05251389	Phase I	ICI refractory melanoma	FMT (by endoscopy)	24
NCT03353402	Phase I	Anti-PD1 refractory melanoma	Oral FTM + Nivolumab	10
NCT03772899	Phase I	Metastatic or unresectable melanoma	Oral FMT + ICI	20 ^1^
NCT04521075	Phase I	Metastatic or unresectable melanoma *	Oral FMT + Nivolumab	42 ^1^
NCT03934827	Phase I	Resectable melanoma ***	MRx0518 vs. placebo prior to surgery	120 ^1^
NCT03817125	Phase I	Metastatic or unresectable melanoma	Nivolumab + SER-401 vs. placebo	10
NCT03819296	Phase I and II	Melanoma ** with ICI-related GI complications	FMT (by endoscopy)	800
NCT04988841	Phase II	Unresectable metastatic or ICI-naïve melanoma	ICIs + MaaT013a enema vs. placebo	60 ^1^
NCT04951583	Phase II	Untreated melanoma *	Nivolumab + ipilimumab + FMT capsules	70 ^1^
NCT04645680	Phase II	Stage III and stage IV melanoma	Diet intervention (isocaloric high fiber vs. whole foods diet) + Immunotherapy	42
NCT04577729	Randomized	PD-1 refractory melanoma	ICIs +FMT vs. FMT	60 ^1^
NCT04866810	Randomized	Unresectable or untreated metastatic melanoma	Anti-PD-1 or anti-PDL-1 + behavioral diet vs. observation	60 ^1^

^1^ Planned enrollment. * Also includes NSCLC. ** Also includes GU and lung cancer. *** Also includes other solid tumors. MaaT013a is a microbiome restoration biotherapeutic. MRx0518 is a formulation of enterococcus species. SER-401: capsules made up of healthy human donors’ purified suspension of firmicute spores.

**Table 4 ijms-24-09702-t004:** Observational studies assessing the interaction between ICIs and melanoma.

NCT	Patient Selection	Treatment	Observation or Test	Primary Outcome	N
NCT05102773	Stage III or IV melanoma	ICIs	Blood and stool samples	Alpha-diversity change	89
NCT04136470	Melanoma *	ICIs	Stool sample	Gut microbial diversity	130
NCT02600143	Melanoma with colitis	ICIs	Stool sample	Gut microbial differences in colitis development	123
NCT04875728	Stage I and II melanoma	Surgery + Cefazolin (surgical prophylaxis)	Stool sample	Change in microbiome after prophylactic antibiotics	20
NCT04698161	Melanoma *	ICIs	Stool, saliva, urine, and blood samples	Microbiome biobank collection	50
NCT0364289	Stage III and IV melanoma	ICIs	Stool and blood samples	Stool microbial diversity, peripheral blood cell immunophenotyping, adverse effects	450
NCT05037825	Malignant Melanoma **	Anti-PD-1, Anti-PDL-1 and anti-CTLA-4 alone or in combination	Stool samples	Change in gut microbiome composition with treatment	800
NCT04734704	ICI treated melanoma ***	Anti-PD-1	Skin Swabs on lesion and non-lesion sites	Skin microbial composition	175
NCT04107168	Stage III and stage IV melanoma	Anti-PD-1 alone or in combination with Anti-CTLA-4	Stool and saliva samples	Gut microbiome’s effect on 1-year PFS	

* Also includes non-small cell lung cancer. ** Also includes NSCLC, RCC, and triple negative breast cancer. *** Also includes patients with vitiligo or those who developed vitiligo after ICI treatment.

**Table 5 ijms-24-09702-t005:** The role of the microbiota in influencing rate of immune mediated adverse events in skin cancers.

Bacteria	Immunotherapy	Adverse Event	Main Findings	Reference
*Staphylococcus epidermidis*	CTLA-4 mAb	Skin inflammation	Mice skin colonized with *S. epidermidis* followed by treatment with systematic ICI developed skin inflammation was seen on days 6 to 8 of treatment. No skin inflammation was seen in treatment of mice with ICI alone or colonization of *S. epidermidis* alone.	[56]
*Bacteroidetes* phylum (*Bacteroidaceae*, *Rikenellaceae*, *Barnesiellaceae)*	CTLA-4 mAb	Colitis	*Bacteroidetes* phylum was associated with increased resistance to colitis.	[60]
*Bacteroidetes* phylum, *Firmicutes*	CTLA-4 mAb	Colitis	*Bacteroidetes* phylum was associated with resistance to colitis. *Firmicutes*-rich microbiota were associated with increased tendency to colitis.	[39]
*Bacteroidetes* phylum, *Firmicutes*	PD-1 mAb + CTLA-4 mAb	Colitis	*Bacteroidetes* phylum was associated with increased tendency to colitis. *Firmicutes*-rich microbiota were associated with resistance to colitis.	
*Streptococcus*, *Paecalibacterium*, *Stenotrophomonas*, *Faecalibacterium*, *Bacteroides*	PD-1 mAb	Pruritis and/or rash, thyroid dysfunction, and diarrhea	Severe irAE: gut microbiota abundant with *Streptococcus* and *Stenotrophomonas* (diarrhea) *Paecalibacterium* (thyroid dysfunction) Mild irAE: gut microbiota abundant with *Faecalibacterium and Bacteroides* (diarrhea)	[64]

## Data Availability

No new data were created or analyzed in this study. Data sharing is not applicable to this article.

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
