# Peer review of "Microbial Influences on Immune Checkpoint Inhibitor Response in Melanoma: The Interplay between Skin and Gut Microbiota"

_ijms, 2023, doi:10.3390/ijms24119702_

Round 1
Reviewer 1 Report
Manuscript ID: [IJMS] Manuscript ID: ijms-2388835 - REVIEW REPORT
Type of manuscript: Review
Title: Microbial Influences on Immune Checkpoint Inhibitor Response in Melanoma: The Interplay between Skin and Gut Microbiota
The authors summerized evidences on the influence of gut and skin microbiota and relative metabolites on Immune Checkpoint Inhibitor Response in Melanoma.
Mechanistic exploration provides novel insights for developing rational microbiota-based therapeutic strategies by manipulating gut microbiota, such as fecal microbiota transplantation (FMT), probiotics, engineered microbiomes, and specific microbial metabolites
COMMENTS
In my opinion the rewiew is well written and the authors did an appreciable work.
I have minor remarks.
1) Immune checkpoint inhibitors are associated with a wide range of immune-related adverse events including musculoskeletal manifestations (Rheumatology International 41, 33–42,2021). This aspect is missing in the review
2) At line 165 please explaine acronym FMT, also used at line 166,181, 182 and explained just at line 220.

Author Response
Thank you for your time and efforts in reviwing our manuscript. Please find below our replies to the comments:
1. Immune checkpoint inhibitors are associated with a wide range of immune-related adverse events including musculoskeletal manifestations (Rheumatology International volume 41, pages33–42,2021). This aspect is missing in the review
Thank you for your comment. This has been added to the manuscript with the above provided citation:
In addition, ICIs have been associated with musculoskeletal adverse events including inflammatory arthritis, myositis and polymyalgia rheumatica.
2. At line 165 please explaine acronym FMT, also used at line 166,181, 182 and explained just at line 220
This has been corrected in the manuscript. FMT was defined only when first encountered in the manuscript and in the abbreviation section.
Reviewer 2 Report
The review by Bouferraa et al. is a comprehensive, well-articulated analysis of the role skin microbiota play in the immune checkpoint response in melanoma. I have no significant issues with the manuscript. However, one point of contention is the lack of differentiation from the authors' previous review in 2021 - Bouferraa Y, et al. "The Role of Gut Microbiota in Overcoming Resistance to Checkpoint Inhibitors in Cancer Patients: Mechanisms and Challenges." Int J Mol Sci. 2021 Jul 27;22(15):8036. doi: 10.3390/ijms22158036. PMID: 34360802; PMCID: PMC8347208. In fact, the authors have even reused Figure 1 from their previous review.
A discussion addressing this concern would be beneficial for readers. I recommend the authors incorporate a section in the manuscript that explicitly illustrates what was reviewed in 2021, what has changed or evolved in the field by 2023, and why they felt a new review was warranted. Once this issue is adequately addressed, the manuscript will stand as an even more significant contribution to the literature on this topic
Author Response
Thank you for your time and effort in reviewing our manuscript. While our previous work in 2021 was a general review about the role of only gut microbiota in the response to ICIs irrespective of the cancer type, this article provides an updated review on the role of both gut and skin microbiota in melanoma in particular. A section has been added to the manuscript’s introduction as suggested. It is highlighted below:
In our previous work published in 2021, we have discussed the role that the gut microbiota plays in overcoming the resistance to ICIs used in the treatment of different cancer types (8). As the field has significantly evolved in the past couple of years, the role of gut microbiota has been extensively studied in many types of cancer, requiring an extensive review for its role in every caner type alone.
In this article, we aim to discuss the potential mechanisms underlying the interaction between the skin and gut microbiota and the immune system in melanoma, the current evidence supporting the role of the microbiota in ICI response, and the future implications for clinical practice. As such, this article will focus on the role of gut as well as skin microbiota in the development and response of melanoma to ICIs, providing a focused update on this interaction in melanoma patients in particular (8).